# Nanomechanics and co-transcriptional folding of Spinach and Mango

Jaba Mitra[1,2] & Taekjip Ha [ID] [2,3,4,5]

Recent advances in fluorogen-binding "light-up" RNA aptamers have enabled protein-free detection of RNA in cells. Detailed biophysical characterization of folding of G-Quadruplex (GQ)-based light-up aptamers such as Spinach, Mango and Corn is still lacking despite the potential implications on their folding and function. In this work we employ single-molecule fluorescence-force spectroscopy to examine mechanical responses of Spinach2, iMangoIII and MangoIV. Spinach2 unfolds in four discrete steps as force is increased to 7 pN and refolds in reciprocal steps upon force relaxation. In contrast, GQ-core unfolding in iMangoIII and MangoIV occurs in one discrete step at forces >10 pN and refolding occurred at lower forces showing hysteresis. Co-transcriptional folding using superhelicases shows reduced misfolding propensity and allowed a folding pathway different from refolding. Under physiologically relevant pico-Newton levels of force, these aptamers may unfold in vivo and subsequently misfold. Understanding of the dynamics of RNA aptamers will aid engineering of improved fluorogenic modules for cellular applications.

[1] Department of Materials Science and Engineering, University of Illinois at Urbana-Champaign, Urbana, IL 61801, USA. [2] Department of Biophysics and Biophysical Chemistry, Johns Hopkins University, Baltimore, MD 21205, USA. [3] Department of Biophysics, Johns Hopkins University, Baltimore, MD 21218, USA. [4] Department of Biomedical Engineering, Johns Hopkins University, Baltimore, MD 21218, USA. [5] Howard Hughes Medical Institute, Baltimore, MD 21218, USA. Correspondence and requests for materials should be addressed to T.H. (email: tjha@jhu.edu)

Non-coding RNAs (ncRNAs), including microRNAs, circular RNAs, long intergenic ncRNAs, and small nucleolar RNAs, regulate many cellular processes underlying gene expression, cell fate determination, morphogenesis, and polarization[1]. Because RNA is inherently nonfluorescent, fluorescent reporter molecules must be added for RNA visualization in live cells. An RNA binding protein fused with a fluorescent protein can be directed to a modified RNA that contains a protein binding motif, and incorporation of dozens of repeats into a gene of interest allows researchers to visualize single messenger RNA molecules with high spatiotemporal resolution in cells[2]. However, this method requires large modifications to the gene and the RNA that may potentially perturb function and can cause high fluorescent background coming from unbound proteins[3]. Since the discovery that an RNA aptamer can specifically interact with a fluorogen to generate a module that makes the fluorogen much brighter than its counterpart free in solution[4], many RNA-fluorogen complexes have been developed and used for fluorescent detection of RNA in cells[5–12]. Such an RNA aptamer that can specifically interact with a fluorogen to elicit fluorescence is known as a "light-up" aptamer[13,14].

The first such light-up aptamer shown to be effective for live-cell RNA imaging is spinach. Spinach spectrally emulates green fluorescent protein (GFP) upon binding to DFHBI, an analog of the GFP chromophore[5]. Although spinach variants such as Spinach2[6,15] have been engineered with improved fluorogenic activities, because of their large size and complex geometry, they are known to misfold and hence hinder imaging applications[6]. A recent addition to the repertoire of light-up aptamers is mango, which specifically binds to a functionalized thiazole orange (TO1)[8,9]. Mango and its variants constitute the smallest light-up aptamers developed to date. A common structural feature of spinach and mango is the presence of a fluorogen-binding G-Quadruplex (GQ) core with mixed parallel and antiparallel connectivity, flanked by A-form duplex stem(s)[16–18]. In addition, non-GQ light-up aptamers that can selectively bind to malachite green[19,20] or cyanine dyes with submicromolar affinity have also been developed[21,22].

RNA GQs have been proposed to play regulatory functions in transcriptional and posttranscriptional processes[23,24], and have been studied in vitro, in bulk solution[25–28] and also at the single-molecule level[29]. Although putative GQ forming sequences have been identified in the transcriptome, the folding propensity of RNA GQs in cells is still under debate[30,31]. The light-up aptamers integrate the distinct structural characteristics of RNA GQs with their ligand-binding ability. Although folding of RNA aptamers found in riboswitches has been studied extensively[32–35], biophysical characterizations of light-up aptamers regarding their folding properties are sparse[36,37]. For example, in order to function efficiently in cells, an RNA aptamer must fold correctly and stably and resist inadvertent unfolding induced by mechanical forces exerted by the fluctuating environments or by RNA binding or RNA-translocating proteins.

In this study, we investigated GQ formation in Spinach2 and mango variants, iMangoIII and MangoIV, at the single-molecule level[38] either in refolding format where folding of a fully synthesized RNA is induced by changes in ionic conditions or in vectorial folding format where nascent RNA is allowed to fold during synthesis. Furthermore, in order to characterize their nanomechanical properties, we performed fluorescence-force spectroscopy using a hybrid instrument combining single-molecule fluorescence resonance energy transfer (smFRET) and optical tweezers[39]. Our results illustrate mechanical superiority of mangos, with respect to Spinach2, making them better adept to physiological levels of tension.

## Results

### Spinach2 folding at the single-molecule level.

We measured the folding properties of Spinach2 at the single-molecule level, first using the refolding format where pre-synthesized RNA is induced to fold through the addition of monovalent or divalent ions. A DNA strand functionalized with biotin at the 5′ end and with Cy5 at the 3′ end and another DNA labeled with Cy3 at the 5′ end were annealed to a 5′ extension of the Spinach2 construct (Fig. 1a, b). FRET efficiency $E$ between the donor (Cy3) and acceptor (Cy5) reflects the end-to-end distance between $A_{24}$ and $A_{75}$ of the Spinach2 sequence[17] and is expected to be higher in the folded state (Fig. 1a–c and Supplementary Fig. 1a)[17]. Structural studies have revealed a RNA GQ at the core of Spinach2[17]. Because GQs are sensitive to monovalent cations[40], we first examined Spinach2 folding as a function of $K^+$ concentration. GQ formation was visible at ~30 mM $K^+$ as suggested by the emergence of a population at $E \sim 0.6$, which culminated in a major population at $E \sim 0.72$ at the physiologically relevant concentration $K^+$ of 100 mM (Fig. 1d (top) and Supplementary Fig. 1c (left)). Circular dichroism (CD) spectroscopy confirmed GQ formation with signatures of parallel loop connectivity (max ~264 nm and min ~235 nm; Supplementary Fig. 1b)[16,17,41]. Although divalent cations such as $Mg^{2+}$ do not significantly affect folding of canonical DNA GQs[40] they can affect fluorogenic activity of spinach in vivo[7,15]. In the presence of 5 mM $Mg^{2+}$ only, Spinach2 adopts a ssRNA-like conformation[42] at $E \sim 0.1$ (Supplementary Fig. 1c). With both $Mg^{2+}$ (5 mM) and $K^+$ (100 mM), CD spectroscopy showed GQ formation (Supplementary Fig. 1b)[41] and the major smFRET population had a peak at a higher $E$ value compared with 100 mM $K^+$ alone (~0.86 vs ~0.72) (Fig. 1d and Supplementary Fig. 1c), suggesting additional compaction induced by $Mg^{2+}$.

Our Spinach2 construct binds to its cognate fluorogen, DFHBI-1T (Fig. 1e–g), and enables GFP-like fluorescence (Supplementary Fig. 1d)[15,43]. Approximately 39% of the Spinach2 molecules tethered on the single-molecule surface manifested GFP-like fluorescence upon addition of DFHBI-1T (Supplementary Fig. 1f). Maturation propensities of wt-GFP and EYFP are respectively ~50% and ~75%, at 32 °C[44]. However, in view of the ultrafast dissociation kinetics of DFHBI from spinach under continuous laser illumination[45] used in our study, a significant population of fluorescent complexes may have been inadvertently missed at our time resolution of 200 ms. Notwithstanding, structural changes, if any, induced upon fluorogen binding could not be resolved via CD spectroscopy (Supplementary Fig. 1e) or smFRET (Fig. 1d (bottom)), consistent with crystallographic studies that showed similarities in global structure between the fluorogen-bound and unbound states[16].

### Spinach2 unravels in a stepwise manner under tension.

For fluorescence-force spectroscopy, the 5′ overhang of a λ-phage DNA was annealed to the dually labeled Spinach2 construct via a short λ-bridge DNA. The other 5′ overhang of λ-phage DNA was annealed to a digoxigenin-labeled DNA strand for subsequent attachment to a micron-sized bead coated with anti-digoxigenin. The bead was optically trapped at a fixed position and a piezo-driven sample stage was translated at a speed of 455 nm s$^{-1}$ to change gradually the force applied to the surface-tethered Spinach2 between ~0.3 and 28 pN. Simultaneously, smFRET time trajectories were recorded (Fig. 2a).

A single molecule of Spinach2 in solution containing 100 mM $K^+$ and 5 mM $Mg^{2+}$ showed stepwise decreases in $E$ value from ~0.86 to a final value of ~0.08 as the force was gradually increased to ~7 pN (Fig. 2b). Upon force relaxation at the same stage moving speed, we observed stepwise increases in $E$ back to the

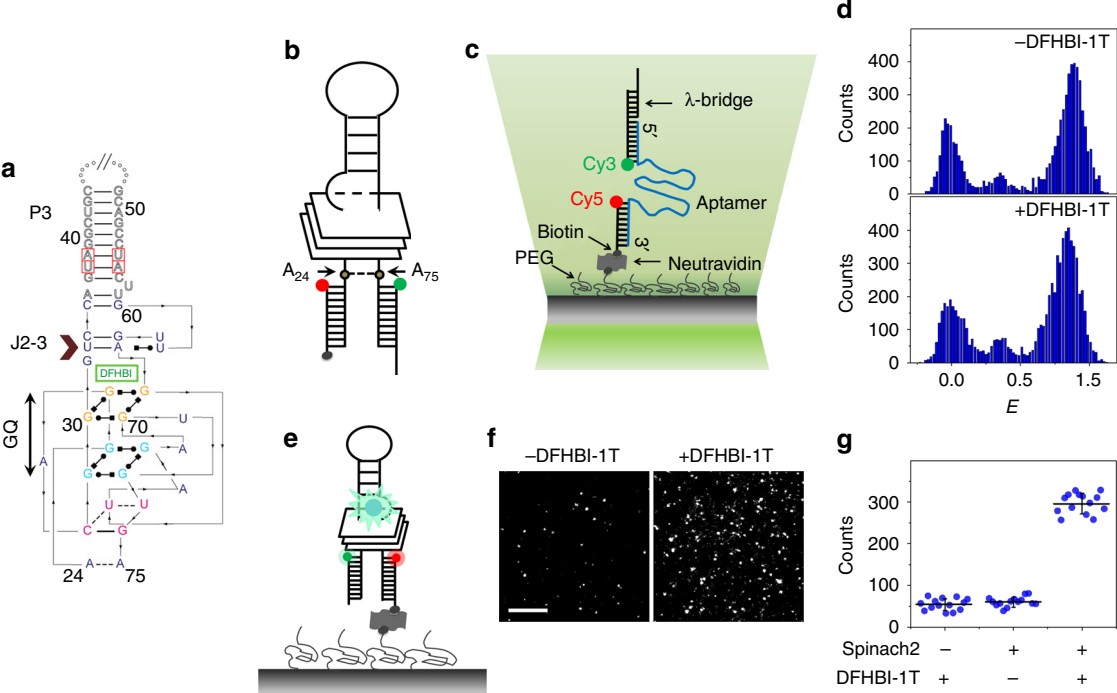

**Fig. 1** Conformational analysis of a Spinach2 aptamer. **a** Sequence and secondary structure of Spinach2-DFHBI used in this study[17]. Adapted with permission from ref. [17]. The Spinach2 GQ, flanking duplex stem, P3, and the base-triple junction J2-3 are indicated in the figure. **b** A schematic of Spinach2 construct used in our studies. The nucleotides are numbered with respect to the Spinach2 crystal structure[17]. The FRET pair of dyes, Cy5 (red) and Cy3 (green) are located adjacent to $A_{24}$ and $A_{75}$ of Spinach2, respectively. Biotinylation of the Cy5-labeled oligo is represented by a black sphere. **c** A schematic representation of the smFRET assay. The RNA aptamer (Spinach2, in this figure) was annealed to Cy3 (donor) and Cy5 (acceptor) labeled handles on the 5′ and 3′ sides, respectively. The Cy5-labeled oligo is biotinylated at the 5′ end. The construct is immobilized on the PEG-passivated surface via neutravidin–biotin interaction. FRET was measured between Cy3 and Cy5. The Cy3-labeled handle is further annealed to 30 nt long λ-bridge for use in fluorescence-force measurements. FRET was measured between Cy3 and Cy5. **d** $E$ histograms of Spinach2 in a buffer supplemented with 100 mM K⁺ and 5 mM Mg²⁺, in the absence (top) and presence (bottom) of DFHBI-1T. **e** Schematic representation of DFHBI-1T (blue sphere) bound Spinach2 on a single-molecule platform. **f** The fluorogenic module elicits GFP-like fluorescence on excitation with 488 nm laser. (Scale bar: 5 µm). **g** Average number of fluorescent spots per image area before and after 30 min incubation with 5 µM DFHBI-1T. Source data are provided as a Source Data File

original value of ~0.86 (Fig. 2b). Stepwise changes in $E$ during both stretching and relaxation were observed across multiple pulling cycles from many single molecules. Remarkably, the $E$ values visited were reproducible enough to show discrete intermediate states even when multiple pulling trajectories were averaged (Fig. 2c). We observed different unfolding intermediate states with $E \sim 0.67$, 0.47, and 0.2 at forces of ~2.5, 4.6, and 7.3 pN, respectively (Fig. 2c, d (top)). Refolding during force relaxation also showed intermediate states with similar $E$ values of ~0.24, 0.45, and 0.67 and at forces of ~6.8, 4.4 and 2.8 pN, respectively (Fig. 2c, d (bottom)). In the presence of saturating concentration of DFHBI-1T fluorogen, Spinach2 unraveled in a similarly stepwise manner but at slightly higher forces (Fig. 2f, g). In view of comparable $E$ values attained during stepwise unfolding and refolding both in the absence and presence of DFHBI-1T, we denote the intermediate states at $E \sim 0.67$, 0.45, and 0.25 as $E_1$, $E_2$, and $E_3$, respectively. Structural studies of spinach by Huang et al.[16] observed local structural changes induced by DFHBI, wherein a new base triple forms atop the fluorogen-binding quadruplex platform (Supplementary Fig. 1a). This may account for the slight improvement in mechanical stability of Spinach2 in the DFHBI-1T bound state.

Cooperative unfolding with a single transition state can be described by the distance to the transition state $\Delta x_u^{\ddagger}$, the average unfolding time at zero-force $\tau_u(0)$, and the apparent free energy of activation $\Delta G^{\ddagger}$[46,47]. DFHBI-1T-binding caused changes in $\Delta x_u^{\ddagger}$s to the $E_1$ ($\Delta x_{u,-DFHBI-1T}^{\ddagger} = 10.1 \pm 1.2$ nm vs $\Delta x_{u,+DFHBI-1T}^{\ddagger} = 8.0 \pm 0.9$ nm) and $E_3$ states ($\Delta x_{u,-DFHBI-1T}^{\ddagger} = 5.8 \pm 0.8$ nm vs $\Delta x_{u,+DFHBI-}$

$_{1T}^{\ddagger} = 4.2 \pm 0.7$ nm), but not the $E_2$ state ($\Delta x_{u,-DFHBI-1T}^{\ddagger} = 6.8 \pm 0.8$ nm vs $\Delta x_{u,+DFHBI-1T}^{\ddagger} = 6.7 \pm 0.8$ nm) (Supplementary Fig. 2a, b). These changes in the $\Delta x_u^{\ddagger}$s may be suggestive of force-induced changes in interactions between DFHBI-1T and Spinach2 motifs during $E_1$ and $E_3$ unfolding but not during $E_2$ unfolding.

Spinach2 in 100 mM K⁺ alone, i.e., without Mg²⁺, also showed stepwise unfolding and refolding (Supplementary Fig. 3a). However, such $E$ steps varied stochastically across multiple pulling cycles of a single molecule and hence could not be unambiguously identified after averaging many $E$ vs force curves (Supplementary Fig. 3b, c). We also noted a significant decrease in fluorescence activation of Spinach2-DFHBI-1T in 100 mM K⁺ alone at the single-molecule level (Supplementary Fig. 4d). Crystallographic studies showed that DFHBI is sandwiched between the top G-quartet and the $U_{32}$-$A_{64}$-$U_{61}$ base-triple junction of the quadruplex flanking stem, P3 (Supplementary Fig. 1a)[16,17]. While canonical G-quartets can form stably in the presence of 100 mM K⁺ only (Supplementary Fig. 1b), Mg²⁺ may be needed to stabilize the junction between the Spinach2 GQ and the flanking stem (Supplementary Fig. 4e)[48]. In fact, single-molecule time trajectories reveal transitions between states with $E \sim 0.7$ and 0.86 upon addition of Mg²⁺ (Supplementary Fig. 4b). The $E \sim 0.7$ state is longer lived at low concentrations of Mg²⁺ (up to ~1.5 mM) but the $E \sim 0.86$ state becomes dominant at 5 mM Mg²⁺ (Supplementary Fig. 4c). Mg²⁺ intercalates between the top G-quartet and the $U_{32}$-$A_{64}$-$U_{61}$ base triple at the junction with the P3 stem[17]. As the P3 stem constitutes the loop connecting Gs across all the G-quartets (Supplementary Fig. 1a),

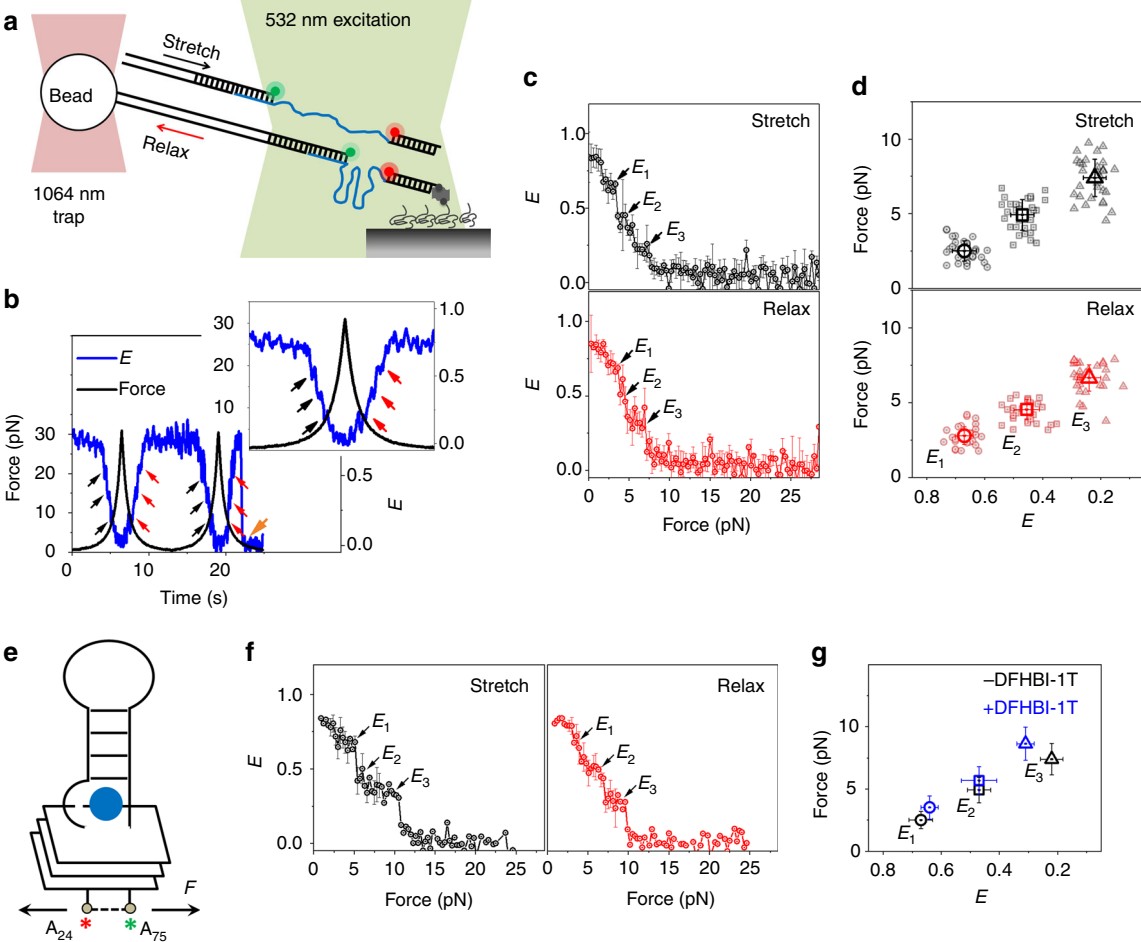

**Fig. 2** Conformational dynamics of Spinach2 under tension. **a** Schematic of integrated fluorescence-force spectroscopy assay: the Spinach2 construct is immobilized on a neutravidin coated quartz surface via a biotinylated strand. The other end is connected to an optically trapped 1 μm diameter bead through a λ-DNA. Force was applied by translating the microscope stage at a speed of 455 nm s$^{-1}$. FRET was measured between Cy3 (donor) and Cy5 (acceptor) as a function of force. **b** A representative time trace of $E$ and the applied force from a single-molecule of Spinach2 over two pulling cycles (20 ms integration time). The black and red arrows represent $E$ steps during stretching and relaxation, respectively. The orange arrow denotes photobleaching of the fluorescent dyes. A blown-up image of cycle 1 is shown in the inset. **c** Average $E$ vs force response from molecules undergoing stretching (top) and relaxation (bottom). The errors represent standard errors. ($N = 48$). **d** Force vs $E$ response corresponding to the steps at $E_1$, $E_2$, and $E_3$ under stretching (top) and relaxation (bottom). The average force and $E$ values are emboldened. Source data are provided as a Source Data File. **e** A schematic of Spinach2 molecule under tension, in the ligand-bound state. DFHBI-1T is represented as a blue sphere. The Cy5 and Cy3 dyes adjacent to $A_{24}$ and $A_{75}$ are shown as red and green asterisks, respectively. Force ($F$) applied on the construct is shown by arrows. **f** Average $E$ vs force response from molecules during stretching (left) and relaxation (right) in the presence of DFHBI-1T. ($N = 36$). **g** Comparison of the average forces stretching forces corresponding to the three unfolding $E$ states in the absence and presence of DFHBI-1T. Source data are provided as a Source Data File

we propose the formation of partially folded GQs ($E \sim 0.7$) in 100 mM K$^+$ alone, which evolve into a stably folded form ($E \sim 0.86$) upon addition of 5 mM Mg$^{2+}$ (Supplementary Fig. 4e). Thus, the $E_1$ state of Spinach2, the first unfolding intermediate under tension in the presence of both K$^+$ and Mg$^{2+}$, may be due to mechanical destabilization of the stem junction and hence partial unfolding of the GQ core.

Overall, our fluorescence-force studies suggest that the Spinach2 GQ is extremely sensitive to forces between 2.5 and 10 pN. Such low force threshold, lower than the forces exerted by RNA motor proteins such as helicases and other RNA binding proteins[49,50], may lead to inadvertent unfolding of the aptamer and compromise fluorogen binding and hence "light-up" properties of Spinach2 in cellular imaging applications[37].

**Conformational dynamics of mango aptamers**. Similar to Spinach2, mango aptamers fold around a GQ core containing both parallel (predominant) and antiparallel connectivity (Fig. 3a)[18].

However, because of the smaller size and simpler geometry compared with spinach, they were suggested to fold more robustly than spinach[18]. Since the first report of engineered mango aptamer by Dolgosheina et al.[8], mango variants with improved fluorescent properties, binding affinities, etc., have been identified via functional reselection[9,51]. In view of the superior fluorogenic properties of $i$MangoIII[51] among the mango aptamers designed till date, we next investigated $i$MangoIII. In the absence of force, K$^+$-induced folding of $i$MangoIII gave a smFRET population at $E \sim 0.55$ (Fig. 3a, c (left) and Supplementary Fig. 5b (left)) with a CD spectral signature corresponding to parallel GQs (Supplementary Fig. 5a)[9,41]. Interestingly, on supplementing the buffer with 5 mM Mg$^{2+}$, we observed an additional structural evolution to a stable state with $E \sim 0.74$ (Fig. 3b, c (right), Supplementary Fig. 5b (right), c). Because the crystallographic structure of $i$MangoIII showed that divalent cations coordinate base triple formation beneath the G-quartet, we attribute the $E$ transition from ~0.55 to ~0.74 to the

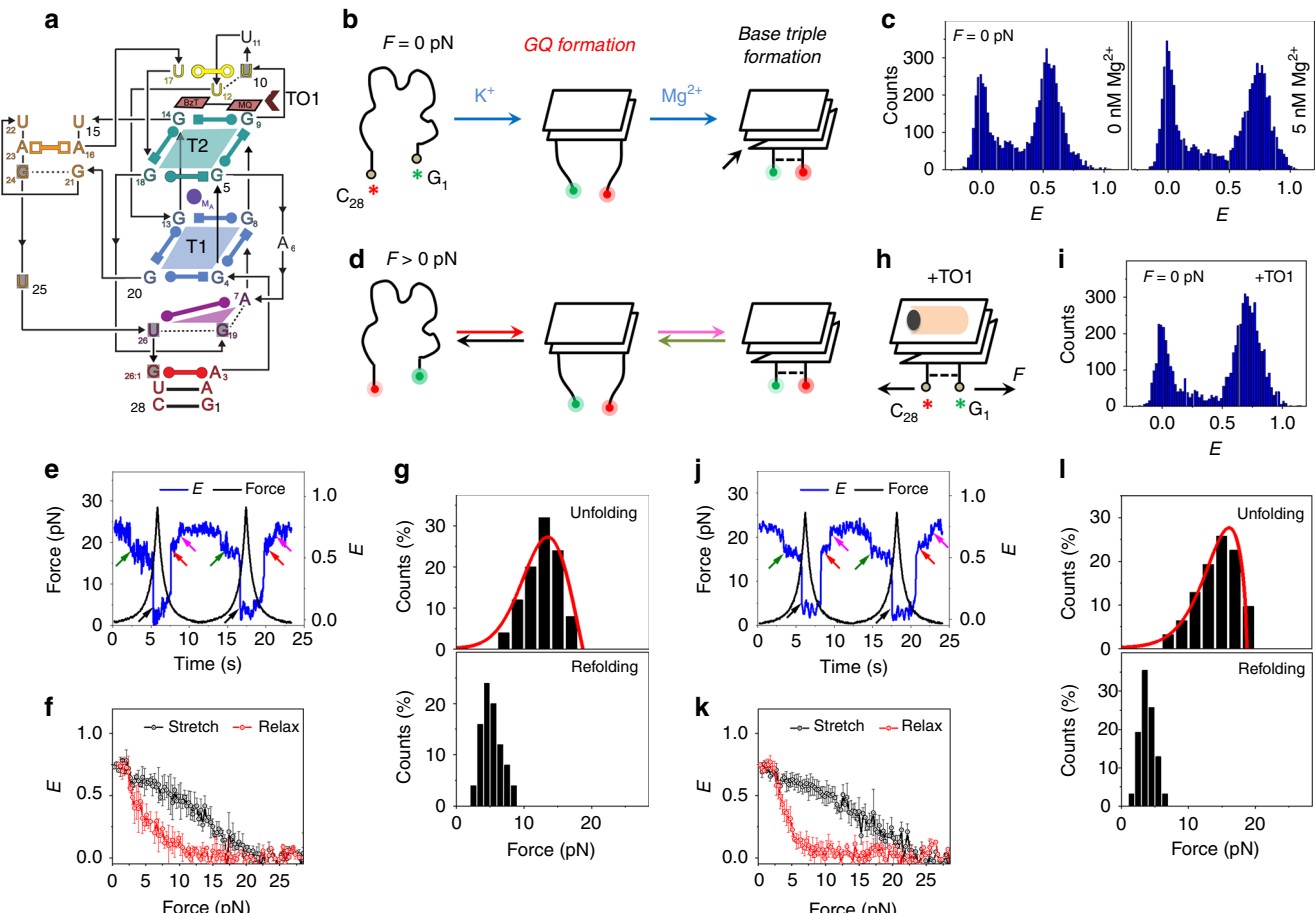

**Fig. 3** Conformational dynamics of *i*MangoIII. **a** Sequence and secondary structure of *i*MangoIII used in this study. T1 and T2 denote the G-quartets in *i*MangoIII. TO1 is indicated by the brown arrowhead[51]. Adapted with permission from ref. [51]. **b** Schematic of probable sequential folding of *i*MangoIII GQ and the flanking base triple, in the absence of force. The structures are drawn with reference to the recently solved crystal structure[51]. The Cy5 and Cy3 dyes adjacent to $C_{28}$ and $G_1$ are shown as red and green asterisks, respectively. **c** *E* histograms of *i*MangoIII in a buffer containing 100 mM $K^+$, with (right) and without (left) $Mg^{2+}$, in the absence of force. **d** Schematic of unfolding (green and black) and refolding (red and magenta) transitions of *i*MangoIII, under an applied force. Green and magenta arrows respectively indicate unfolding and refolding of the base triple. Black and red arrows indicate unfolding and refolding of the GQ core, respectively. **e** A representative smFRET time trajectory over two pulling cycles. Unfolding and refolding transitions at ~2.5 pN are indicated by green (stretch) and magenta (relax) arrows. The major unfolding and refolding transitions are shown via black and red arrows, respectively. **f** Average *E* vs force response. **g** Distributions of unfolding and refolding forces corresponding to the black and red arrows in **d**. (*N* = 62 for both). The red curves represent force distributions estimated from the Dudko–Szabo model[46,47] (top). **h** A schematic of TO1-*i*MangoIII complex under tension. TO1 is represented by the orange and black symbol. Force (*F*) applied on the construct is shown by arrows. **i** *E* histogram of *i*MangoIII under saturated concentrations of TO1, in the absence of force. **j** A representative smFRET time trajectory of *i*MangoIII in the presence of TO1. The arrow indicators used are similar to **e**. **k** Average *E* vs force response in TO1 containing buffer. **l** Distributions of unfolding and refolding forces corresponding to the black and red transitions in **i** and fits (red curves) predicted from models proposed by Dudko et al.[46,47] (*N* = 50 for both). Data presented in **e–l** were collected in a buffer containing 100 mM $K^+$ and 5 mM $Mg^{2+}$. The time trajectories were acquired at an integration time of 20 ms. All error bars represent standard errors

stacking between $A_7$, $G_{19}$, and $U_{26}$ (Fig. 3a, b)[51]. This is similar to the $Mg^{2+}$-induced stabilization of base triple at the junction between GQ and a flanking stem observed in Spinach2.

During a pulling cycle in 100 mM $K^+$, *i*MangoIII typically underwent gradual decreases in *E* from ~0.55 to ~0.4, followed by an abrupt transition to *E* ~ 0.05 (Supplementary Fig. 6b). Upon relaxation, we observed an abrupt increase in *E* followed by a gradual increase back to *E* ~ 0.55 (Supplementary Fig. 6b). The unfolding ($f_{unfold}$) and refolding ($f_{refold}$) forces were defined as those corresponding to the midpoint of transitions between *E* ~ 0.4 and ~ 0.05 ($f_{unfold}$) and vice versa ($f_{refold}$). Such transitions are suggestive of cooperative unfolding, similar to that of canonical GQs[52]. In all, unfolding occurred at a range of force values with a peak at ~12 pN, while refolding occurred at ~3.5 pN showing hysteresis (Supplementary Fig. 6c, d). The gradual *E* changes

prior to an abrupt cooperative transition are likely due to stretching of the ssRNA overhang, which is yet to be incorporated into the A-G-U base triple in the absence of magnesium (Supplementary Fig. 6b, c).

In contrast, in the presence of both $K^+$ and $Mg^{2+}$, *i*MangoIII displayed two discrete steps of unfolding. A typical fluorescence-force response was characterized by an abrupt *E* transition from ~0.74 to ~0.6 at ~2.5 pN (Supplementary Fig. 7 (top, left)), followed by an abrupt transition to ~0.05 at ~13 pN (Fig. 3d–f). Similar two-step features were observed during refolding, first to *E* ~ 0.65 at ~4.5 pN and then to *E* ~ 0.74 at 2.5 pN (Fig. 3d–f and Supplementary Fig. 7 (bottom, left)). The $\Delta x_u^{\ddagger}$ value we estimated for the second stage of unfolding was $3.8 \pm 0.7$ nm ($f_{unfold}$ ~ 12 pN) and was similar to the $\Delta x_u^{\ddagger}$ value of $4.2 \pm 0.5$ nm from the single unfolding transition observed in 100 mM $K^+$

without Mg$^{2+}$, suggesting that both share a common GQ core. The addition of Mg$^{2+}$ induces an auxiliary base triple to stabilize the overall structure. The base triple unfolds at ~2.5 pN, with a corresponding $\Delta x_u^{\ddagger}$ of 7.2 ± 1.8 nm (Supplementary Fig. 7). We observed ssRNA-like behavior with 5 mM Mg$^{2+}$ and no K$^+$ (Supplementary Fig. 9a, d, e)[42,53].

Upon binding the cognate fluorogen, polyethylene-glycol (PEG)-functionalized biotinylated thiazole orange (henceforth referred to as TO1), $i$MangoIII showed GFP-like emission at ~530 nm (Supplementary Fig. 8a)[9]. As in Spinach2, no global change in the underlying GQ topology could be detected via CD (Supplementary Fig. 8b) or smFRET (Fig. 3i and Supplementary Fig. 8c, d). We further examined $i$MangoIII binding to TO1, immobilized on the imaging surface through its biotin, by imaging Cy3 attached to $i$MangoIII (Supplementary Fig. 8e, f). We observed much reduced binding of $i$MangoIII to TO1 in the absence of K$^+$ (buffer containing 50 mM Tris only or 5 mM Mg$^{2+}$), likely because the GQ core is necessary for the aptamer to bind the fluorogen (Supplementary Figs. 8a, f and 9b). $i$MangoIII binding to TO1 improved by an additional 1.3-fold when 5 mM Mg$^{2+}$ was added to 100 mM K$^+$ buffer (Supplementary Fig. 8a, f).

$i$MangoIII unraveled in two discrete steps also in the presence of TO1 (Fig. 3j, k). The first step due to unfolding of the base triple occurred at ~2.7 pN, with a $\Delta x_u^{\ddagger}$ of ~6.8 ± 1.8 nm (Supplementary Fig. 7). $f_{unfold}$ and $\Delta x_u^{\ddagger}$ values are similar with and without TO1, consistent with lack of direct interaction between TO1 and the base triple in the crystal structure (Supplementary Fig. 7)[51]. For the second step of unfolding, average $f_{unfold}$ and $f_{refold}$ were ~15 and 3.5 pN, respectively (Fig. 3l), close to that observed in the absence of TO1 but $\Delta x_u^{\ddagger}$ decreased from 3.8 and 2.8 nm upon TO1 addition, suggesting TO1 binding imparts structural changes.

We next examined MangoIV, which is predicted to have a simpler, three-tiered GQ core, similar to the original mango I but unlike $i$MangoIII which has a two-tiered GQ core[9]. MangoIV showed K$^+$-dependent folding (Supplementary Fig. 10c), and adopted a parallel conformation in 100 mM K$^+$ (Supplementary Fig. 10b)[41] and a smFRET population at $E \sim 0.85$ (Supplementary Fig. 11a). Under increasing tension, MangoIV maintained a stable $E \sim 0.85$ signal until an abrupt one-step transition to $E \sim 0.05$. Refolding also occurred in a single step but at lower forces (Supplementary Fig. 11b, c). We estimated $\Delta x_u^{\ddagger}$ of 2.8 ± 0.6 nm at $f_{unfold}$ of ~18 pN (Supplementary Fig. 11c). Unlike $i$MangoIII, we could not detect gradual decrease in $E$, characteristic of stretching of an ssRNA overhang, preceding the abrupt unfolding of MangoIV. MangoIV is predicted to harbor an 8 bp stem flanking the GQ core[9]. The abrupt, single-step unfolding transition from $E \sim 0.85$ to ~0.05 indicates cooperative destabilization of the stem and the GQ core.

Although the above zero-force smFRET and fluorescence-force spectroscopy measurements indicated cooperative folding of the GQ core and duplex stem of MangoIV in a buffer containing 100 mM K$^+$ only, we repeated the measurements in a buffer containing 1 mM Mg$^{2+}$ in addition to 100 mM K$^+$ to better mimic the intracellular ionic conditions. MangoIV maintained a parallel GQ-like CD spectral signature (Supplementary Fig. 10b)[9] and mechanical responses that are comparable to those observed in 100 mM K$^+$ alone (Fig. 4c–e). Upon addition of TO1, MangoIV showed GFP-like fluorescence with an emission maximum at ~535 nm (Supplementary Fig. 12a)[9] but did not exhibit global changes in structure detectable via CD (Supplementary Fig. 12b) or smFRET (Fig. 4g and Supplementary Fig. 12d). Under saturating concentrations of TO1, MangoIV unfolded at ~17 pN, similar to that obtained without TO1 and $\Delta x_u^{\ddagger}$ values were also similar (2.2 ± 0.5 vs 3.1 ± 0.3 nm with and without ligand,

respectively). However, refolding occurred at ~4 pN, which is lower than ~7.5 pN in the absence of TO1 (Fig. 4h–j).

$i$MangoIII and MangoIV are similar with respect to their fluorogenic properties. However, MangoIV was predicted to have a three-tiered GQ topology, in contrast to the two-tiered GQ structure of $i$MangoIII. The improved mechanical stability of MangoIV ($f_{unfold} \sim 18$ pN) when compared with $i$MangoIII ($f_{refold} \sim 12$ pN) may be attributed to an additional G-quartet[9,51]. Furthermore, unlike $i$MangoIII (and Spinach2), MangoIV folding is insensitive to Mg$^{2+}$, likely because of the absence of a base triplet junction at the interface of MangoIV GQ and the flanking duplex stem[48].

The class of mango aptamers is smaller in size than the spinach aptamers. For example, a minimal spinach construct known as baby spinach comprises of 51 nucleotides[10], whereas mango aptamers consist of an invariant ligand-binding core spanning only 23 nucleotides[9]. Our fluorescence-force measurements demonstrate higher mechanical stability of mango and hence less susceptibility to mechanical forces an RNA molecule may experience in vivo during its synthesis, processing, trafficking, and function.

In general, stable folding of GQ core in mango was independent of Mg$^{2+}$. Mg$^{2+}$ promotes formation of a base triple underneath the G-quartets in $i$MangoIII without globally affecting the fluorogen-binding GQ core. In contrast, Mg$^{2+}$ directly impacts the GQ core in Spinach2 by stabilizing the connecting stem loop, P3 (Fig. 1a and Supplementary Fig. 1a) at its interface with the GQ. As stable Spinach2 folding entails ~5 mM Mg$^{2+}$, in vivo applications of Spinach2 are likely to be affected by low physiological concentrations of free Mg$^{2+}$ (1–2 mM)[54].

**Vectorial folding of the RNA aptamers.** Accurate folding of the GQ in an RNA aptamer governs its "light-up" functions in vivo. For cellular imaging, an aptamer is fused to and expressed with a target RNA[8]. In cells, transcription may prompt inadvertent GQ misfolding from the 5′ end[55], inhibiting subsequent fluorogen binding[37]. In order to gain insight into co-transcriptional folding of "light-up" aptamers, we used a highly processive superhelicase Rep-X to mimic the vectorial nature of co-transcriptional RNA folding[56,57]. To this end, Rep-X was first loaded on to the surface-immobilized aptamer-heteroduplex where the aptamer portion is annealed to a DNA strand that has a dT$_{20}$ 3′ overhang. Subsequent unwinding of the heteroduplex and RNA folding was initiated by adding Mg$^{2+}$ and ATP (Fig. 5a). As Rep-X translocates along the complementary strand in the 3′ to 5′ direction, the RNA strand is revealed to solution in the 5′ to 3′ direction, which is the direction of transcription, and at a speed matching the transcription speed[57].

ATP was washed away after a minute to capture the maiden folded state of RNA (Fig. 5a). Prior to Rep-X unwinding, smFRET peak between $E \sim 0$ and 0.05 was observed for the aptamer heteroduplexs (Fig. 5b). Rep-X unwinding yielded ~32, 28, and 32% high FRET folded populations centered at $E \sim 0.85$, 0.72, and ~0.87 in Spinach2, $i$MangoIII, and MangoIV, respectively (Fig. 5b). In contrast, the same reaction performed with slowly hydrolyzable ATP analog, AMP-PNP, yielded ≤6% folded populations.

Real-time observations of the surface-tethered molecules informed us of folding intermediates. For example, Spinach2-heteroduplex gradually transitioned from $E \sim 0.02$ to ~0.85 upon unwinding by Rep-X. The gradual increase in $E$ suggests coiling up of the nascent RNA segment freed from the heteroduplex during unwinding prior to folding into Spinach2 GQ (Fig. 5c (left)). In stark contrast, $i$MangoIII folding occurs in two discrete

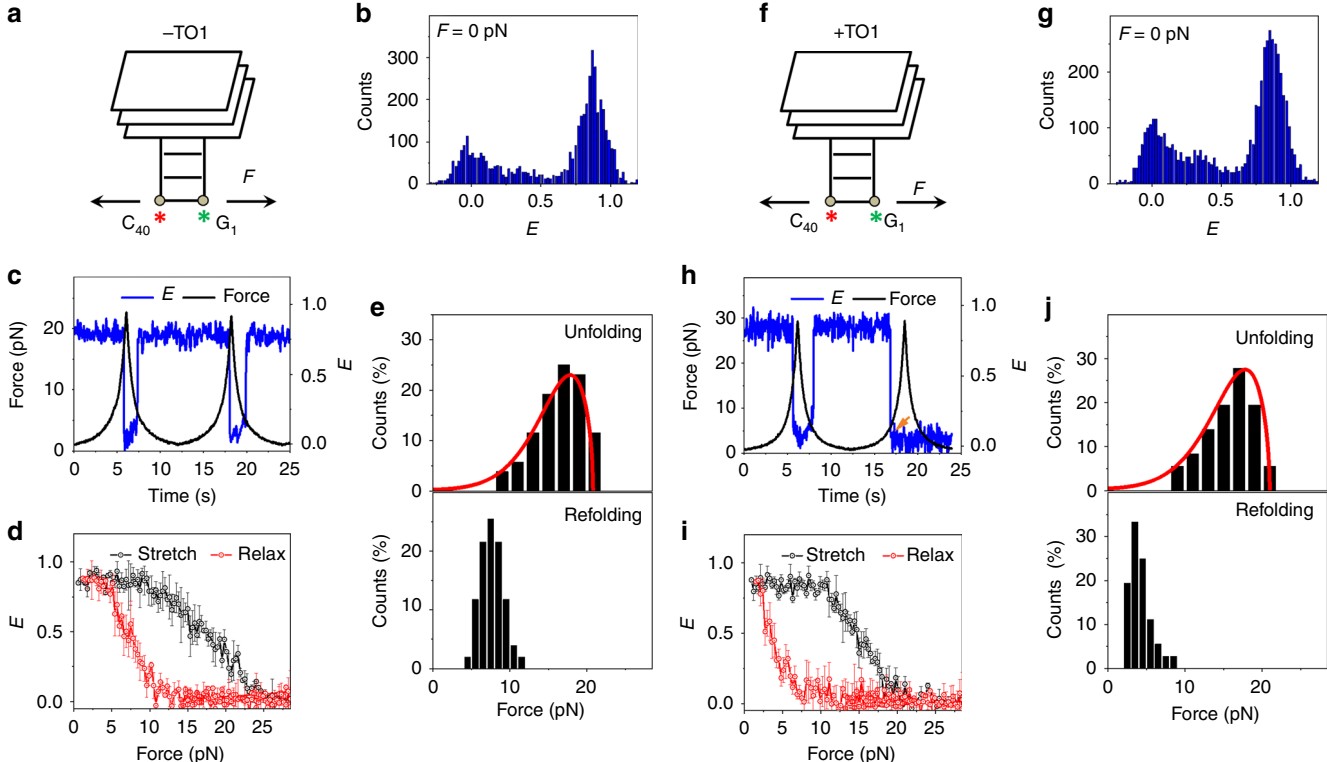

**Fig. 4** Conformational dynamics of MangoIV. **a** A schematic of MangoIV under tension. In view of the proposed structural similarity, the representation is constructed with reference to MangoI crystal structure[9, 18]. The Cy5 and Cy3 dyes adjacent to $C_{40}$ and $G_1$ are shown as red and green asterisks, respectively. To mimic the intracellular ionic conditions, all data shown in this figure were collected in a buffer containing 100 mM $K^+$ and 1 mM $Mg^{2+}$. **b** $E$ histograms of MangoIV, in the absence of force. **c** A representative FRET time trajectory of a single molecule of MangoIV over two pulling cycles. **d** Average $E$ vs force response. **e** Distributions of unfolding (top) and refolding forces (bottom). ($N = 51$ for both). **f** A schematic of TO1-MangoIV module under tension. Force ($F$) applied on the construct is shown by arrows. **g** $E$ histogram of MangoIV in a buffer containing saturated concentrations of TO1, in the absence of force. **h** A representative smFRET time trajectory of MangoIV in the presence of TO1. The orange arrow indicates photobleaching of the dyes. **i** Average $E$ vs force response in TO1 containing buffer. **j** Distributions of unfolding and refolding forces. ($N = 36$ for both). The time trajectories were acquired at an integration time of 20 ms. The red curves in **e** and **j** represent force distributions estimated from the Dudko–Szabo models[46, 47]. All error bars represent standard errors

steps at $E \sim 0.6$ and $\sim 0.72$. We attribute this to sequential formation of the two-tiered GQ core and base triple flanking the GQ, in that order (Fig. 5c (middle), Supplementary Fig. 14a, b). GQs were formed within $\sim 0.4$ s of initiation of duplex unwinding and the base triple formed $\sim 1.9$ s after GQ formation (Supplementary Fig. 14b). Similar $E$ steps observed in MangoIV at $\sim 0.55$ and $\sim 0.87$ can be hypothesized as GQ formation (GQ formation time, $\Delta t_{avg} \sim 0.56$ s) and zipping up of the adjacent stem after $\sim 0.15$ s (Fig. 5c (right) and Supplementary Fig. 14c)[9,18]. Such an intermediate was not observed during unfolding and folding under tension, suggesting that when both GQ and stem elements are available, they fold and unfold cooperatively. Only during "co-transcriptional" folding do we see them as independent events. We note that unlike iMangoIII and MangoIV, proper integration of the G-quartets in Spinach2 is possible only after the release and formation of the flanking A-form stem[16,17]. This may explain the lack of modular folding pattern observed in Spinach2 during vectorial folding.

Ion-induced refolding of the RNA aptamers in vitro showed minor smFRET populations at mid-$E$ (Supplementary Fig. 15a, d). Mechanical response of that minor population is similar to that of single stranded nucleic acids (Supplementary Fig. 15b, c, e)[4] so we attribute that population to misfolded or unfolded RNA. Interestingly, equilibrium distribution of the aptamer populations following vectorial folding lacked such mid-$E$ populations. Sequential assembly of the GQ and the flanking structures in

their nascent states in spinach and mango may have reduced misfolding during vectorial folding[58].

## Discussion

Up until now, the development of a fluorogenic platform has been guided by specificity and affinity of the fluorogen to the RNA aptamer and the resulting fluorescent enhancement and photostability of the module[59]. As the fundamental cellular processes generate and are regulated by pico-Newton levels of tension, mechanical stability of these aptamers is potentially another important consideration. In this study, we canvassed the conformational dynamics of GQ-based RNA aptamers under tension. Our results highlight weak mechanical stability of Spinach2 compared with mangos, which may make it more susceptible to unfolding under in vivo conditions. Our superhelicase-based co-transcriptional folding experiments suggest that folding of these aptamers during transcription would minimize misfolding. However, ion-induced folding experiments showed an increase in misfolded fraction. If spinach indeed unfolds frequently in vivo due to its low mechanical stability, even though it may fold correctly during transcription, it may transit to a misfolded state after mechanically induced unfolding, potentially compromising its utility as a fluorescent tag. However, mechanical instability of spinach may facilitate efficient degradation of spinach-tagged mRNA, thereby rendering it a potentially useful

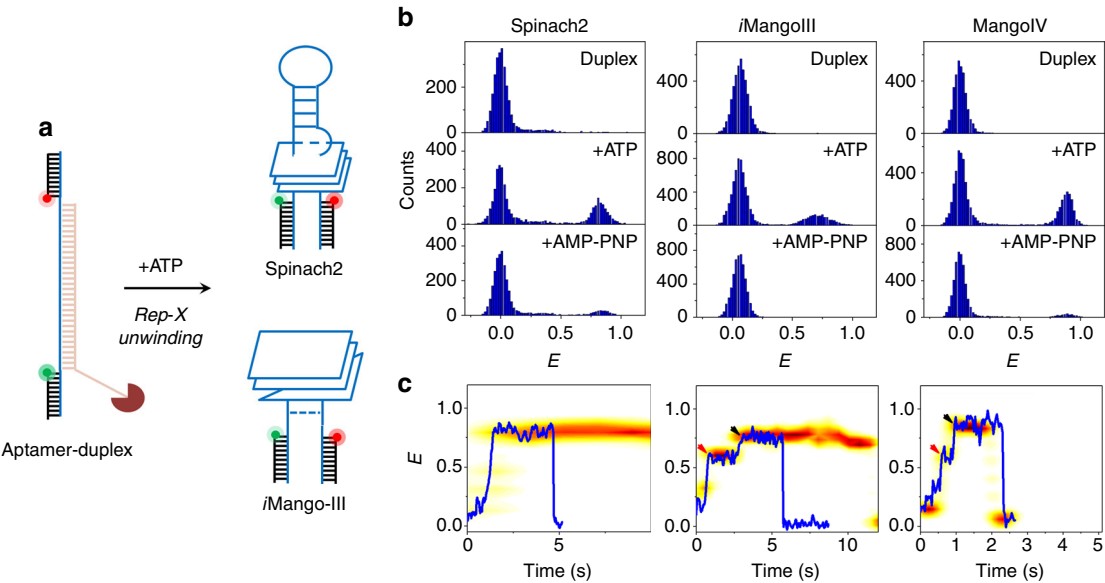

**Fig. 5** Vectorial folding of "light-up" RNA aptamers. **a** Schematic of unfolding of an aptamer-heteroduplex via Rep-X and its subsequent folding (Spinach2 and *i*MangoIII are shown as representations here). **b** *E* histograms of heteroduplex in the Rep-X unbound (top) and after Rep-X unwinding in the presence of ATP (middle) and AMP-PNP (bottom). **c** Representative single-molecule trajectories (blue) showing co-transcriptional folding of an aptamer (Spinach2 (left), *i*MangoIII (middle), and MangoIV (right), 30 ms integration time). The red and black arrows indicate two-step RNA folding corresponding to formation of GQ (red) and the underlying base triple (black, *i*MangoIII) and duplex stem (black, MangoIV). Individual time trajectories have been synchronized ("Methods" section) and overlaid on to the representative traces. Population information is represented as: light yellow (lowest occupancy) to red (highest occupancy)

tool for accurately quantifying the copy number of mRNA with short life spans[60].

Both Spinach2 and *i*MangoIII showed multiple discrete steps in folding and unfolding and we attributed the first steps in their unfolding to base triplets stabilized by $Mg^{2+}$. MangoIV showed a single-step unfolding and folding under tension, which we attribute to a cooperative dissolution of their GQ core and flanking stem. Finally, we note that repeats of these light-up aptamers have been engineered to make a single RNA brighter but in general, the improvement in brightness did not scale up linearly with the number of repeats, likely due to inter-aptamer misfolding[61]. Therefore, the mechanical stability of tandem repeats of aptamers is a topic of interest for future investigations.

## Methods

**DNA constructs**. All RNA and DNA oligonucleotides were purchased from Integrated DNA Technologies. The "light-up" RNA aptamer strands used in this study are as follows:

Spinach2:
5′/gggcggcgaccuAGGACGGGUCCAGUAGGCUGCUUCGGCAGCCUACUUGUUGAGUAGAGUGUGAgccucgcgccgucgcca/3′

*i*MangoIII:
5′/gggcggcgaccuGAAGGAAGGUUUGGUAUGGGGUAGUUGUCGccucgcugccgucgcca/3′

MangoIV:
5′/gggcggcgaccuGGCACGUACCGAGGGAGUGGUGAGGAUGAGGCGAGUACGUGCGccucgcugccgucgcca/3′.

The "light-up" aptamer motif is in upper case. Two complementary stem strands of sequence 5′/Biotin/CGGAGCGACGGCAGCGGT/Cy5/3′ and 5′/Cy3/AGGTCGCCGCCCTCGGGAGCGGACGCACGG were employed as handles to the aptamers for smFRET measurements. Another oligonucleotide strand, λ-bridge of sequence 5′/GGGCGGCGACCTCCGTGCGTCCGCTCCCGA/5′ was used to bridge the aptamer construct with λ-DNA. The aptamer strands were first annealed with the Cy5 and Cy3-labeled complementary strands in 1:1.1(aptamer):1.4 ratio respectively in a buffer containing 50 mM NaCl and 10 mM Tris-HCl, pH 7.5, at 95 °C for 5 min, followed by slow cooling for 3 h and quenching on ice for 5 min. The λ-bridge was further added to the above mixture in the ratio of 2:1 to the Cy5-labeled strand and incubated with rotation at room temperature for an hour. The G4-construct(s) thus generated were used for single-molecule experiments by total internal reflection fluorescence (TIRF) microscopy[38].

For integrated smFRET-optical tweezers assay, the construct generated above was annealed to λ-DNA (New England Biolabs) and the dig-labeled strand. For annealing, λ-DNA (16 nM) was first heated in the presence of 120 mM NaCl at 80 °C for 10 min and then placed on ice for 5 min. The RNA aptamer construct(s) and BSA were added to the λ-DNA at a final concentration of 8 nM and 0.1 mg mL⁻¹, respectively. The mixture was incubated/rotated at room temperature for 2–3 h. Finally, the dig-strand (5′/dig/CCCGCCGCTGGA/dig/3′) was added to a concentration of 200 nM and then incubated with rotation at room temperature for 1 h.

**Sample assembly for smFRET**. The quartz slides and coverslips, passivated with PEG (a mixture of mPEG-SVA and biotin-PEG-SVA, Laysan Bio) were assembled to form imaging chambers[38]. For TIRF experiments, 30 pM RNA aptamer construct(s) were immobilized on the surface via biotin–neutravidin interaction. Finally, imaging buffer was added for data acquisition. Unless otherwise mentioned, the imaging buffer comprised of 50 mM Tris-HCl pH 7.5, 0.8% w/v D-glucose [Sigma], 165 U mL⁻¹ glucose oxidase [Sigma], 2170 U mL⁻¹ catalase [Roche], 3 mM Trolox [Sigma], and predetermined amount of KCl/MgCl₂.

For integrated fluorescence-force measurements, the imaging chamber was incubated in blocking buffer (10 mM Tris-HCl pH 7.5, 50 mM NaCl, 1 mg mL⁻¹ BSA [NEB], 1 mg mL⁻¹ tRNA [Ambion]) for 1 h. The aptamer constructs were then diluted to 10 pM and immobilized on the surface via biotin–neutravidin interaction. Subsequently, 1 μM anti-digoxigenin coated polystyrene beads (Polysciences), diluted in a buffer containing 10 mM Tris-HCl pH 7.5 and 50 mM NaCl, were added to the imaging chamber and incubated for 30 min. Finally, data were acquired in the imaging buffer.

**Fluorescence-force spectroscopy**. An integrated fluorescence-optical trap instrument was recently developed in our lab to study conformational changes of biomolecular systems under tension[39,62]. Briefly, an optical trap was formed by an infrared laser (1064 nm, Spectra-Physics) through the back port of the microscope (Olympus) on the sample plane with a 100× immersion objective (Olympus). A piezo stage with the microscope slide was translated at a speed of 455 nm s⁻¹, between 14 and 16.8–17 μm, over ~6.5 s, in order to apply ~0.3–28 pN force on the sample tethers. The applied force was read out via position detection of the tethered beads using a quadrant photodiode (Thorlabs)[39,62]. A confocal excitation laser (532 nm, World StarTech) was focused on the sample through the side port of the microscope. A piezo-controlled steering mirror (Physik Instrument) raster scanned the sample with the excitation laser. The fluorescence emission was filtered from infrared laser by a band pass filter (HQ580/60 m, Chroma) and excitation by a dichroic mirror (HQ680/60 m, Chroma) and subsequently detected by two avalanche photodiodes (Excelitas Technologies).

**Data acquisition**. For smFRET imaging in the absence of force, we used prism-type TIRF microscopy, with 532 nm laser excitation and back-illuminated electron-multiplying charge-coupled device camera (iXON, Andor Technology)[38]. The smFRET efficiency, $E$, was estimated using $I_A/(I_A + I_D)$, where $I_A$ and $I_D$ are the donor and acceptor intensities, respectively, after background subtraction and crosstalk correction. FRET efficiency histograms were constructed by averaging the first ten data points of each molecule's time trace, acquired at an integration time of 30 ms.

Saturated concentrations of cognate fluorogens (DFHBI-1T (Lucerna): 5 μM[6], TO1 (ABM Inc.): 50 nM[9]) were added to the imaging buffer for understanding the conformational dynamics of fluorogen-bound aptamers. As TO1 is functionalized with biotin, the free neutravidin on the single-molecule surface was concealed with saturated concentrations of biotin (Sigma-Aldrich), prior to addition of the fluorogen. The Spinach2-DFHBI-1T binding was quantified by incubation of 5 μM DFHBI-1T for 30 min followed by imaging with a 488 nm excitation laser (Coherent) and band pass emission filter (HQ 535/30, Chroma Technology). For the mango aptamers, TO1 was pulled down via biotin–neutravidin interaction and Cy3-labeled mango binding was quantified with a 532 nm laser (Cobolt), following the protocol of Dologoshiena et al.[8].

A detailed data acquisition procedure for single-molecule fluorescence-force spectroscopy has been described previously by Hohng et al.[62]. In summary, a tethered bead was trapped and its origin was determined by stretching the tether in opposite directions along $x$- and $y$-axes. The trapped bead was then moved from its origin by 14 μm and a confocal laser was used to scan and locate the fluorescence spot on the tether. Fluorescence emission from the tether molecule was detected concurrent to the application of force, 20 ms after each step in the stage movement.

The unfolding force histograms were analyzed and the free energy parameters such as the transition distances to unfolding and refolding, etc., were predicted using the Dudko–Szabo models ($v = 1/2$)[46,47]. These derived parameters ($\Delta x_u^\ddagger$, $\tau_u(0)$, etc.) were then used to reconstruct the force profiles.

**Vectorial unwinding via Rep-X**. The RNA aptamer construct(s) engineered for smFRET studies was annealed to its complementary sequence with a dT$_{20}$ tail at the 3′ end, in the ratio of 1:1.2 at 95 °C for 5 min, in a buffer containing 50 mM NaCl and 10 mM Tris-HCl, pH 7.5, followed by slow cooling to room temperature. The duplexes were then immobilized on the PEG-passivated quartz slide. Fifty nanomolar Rep-X was then incubated in a loading buffer (10 mM Tris-HCl, pH 7.5, 10% glycerol, 1% BSA) for 2 min. The unbound Rep-X was washed off simultaneously and the unwinding reaction was initiated by adding the unwinding buffer (50 mM Tris-HCl, pH 7.5, 100 mM KCl, 5 mM MgCl$_2$, 1 mM ATP [Thermo Fisher Scientific], 10% glycerol, 1% BSA). The unwinding reaction was quenched after a minute to capture the conformation of maiden folded RNA aptamers. For the real-time measurements, imaging was started a few seconds before addition of the unwinding buffer. A detailed protocol has been described by Hua et al.[57]. The single-molecule time trajectories were aligned based on the PIFE peak centers and the GQ formation and dwell times were determined as shown in Supplementary Fig. 14b. The average GQ formation and dwell times were estimated by fits to Gamma distributions.

**CD Spectroscopy**. CD spectroscopy of the RNA aptamers were performed on an Aviv-420 spectropolarimeter (Lakewood, NJ, USA), using a quartz cell of 1 mm optical path length. The oligonucleotides were diluted to 5 μM in a buffer containing 50 mM Tris pH 7.5 and appropriate concentration of K$^+$/Mg$^{2+}$ ions. The CD spectra was averaged from three scans, recorded between 220 and 320 nm at room temperature and were corrected for baseline and signal contributions from the buffer.

**Fluorescence spectroscopy of "light-up" RNA modules via fluorometer**. Fluorescence measurements of the RNA aptamer and fluorogen complex were performed with a fluorometer (Cary Eclipse Fluorescence Spectrophotometer, Agilent Technologies). Fluorescence spectra were collected under the following conditions: (1) 100 nM Spinach2 and 1 μM DFHBI-1T, $\lambda_{ex} = 482$ nm[6] and (2) 100 nM mango and 200 nM TO1, $\lambda_{ex} = 506$ nm (iMangoIII)[51] and 510 nm (MangoIV)[9]. All measurements were done in a buffer containing 50 mM Tris pH 7.5, 100 mM K$^+$ and 5 mM MgCl$_2$ (1 mM MgCl$_2$ for MangoIV) at room temperature.

**Reporting summary**. Further information on research design is available in the Nature Research Reporting Summary linked to this article.

## Data availability

All relevant data supporting the findings of this study are available within the paper and its supplementary information files. The data underlying Figs. 1g and 2d, g, and Supplementary Figs. 4d, 8f, 9b, and 12f are provided as a source data file. All data are available from the corresponding author upon reasonable request.

## Code availability

The codes used from the analysis of data can be downloaded from http://ha.med.jhmi.edu/resources/.

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

## Acknowledgements
We thank Momčilo Gavrilov for providing Rep-X. This work was supported by the National Science Foundation Grant PHY-1430124 (to T.H.) and the National Institutes of Health Grants GM122569 (to T.H.). T.H. is an Investigator with the Howard Hughes Medical Institute. We thank Dr. David Rueda for providing sequence information of MangoIV before publication. We thank Dr. Adrian Ferré-D'Amaré for sharing sequence information of *i*MangoIII before publication.

## Author contributions
J.M. and T.H. designed the experiments. J.M. performed the experiments and analyzed data. J.M. and T.H. prepared the manuscript.

## Additional information

**Competing interests:** The authors declare no competing interests.

