## [Peer Review File · Nature Communications]

Reviewers' comments:

Reviewer #1 (Remarks to the Author):

Mitra and Ha have carried out a thorough and well-conducted study of two different types of RNA aptamer. They study folding of the aptamers using smFRET and through creative experiments using a combined fluorescence-force setup and a "co-transcriptional-like" system. The results are interesting and provide food for thought for the chemical biology community interested in developing these RNA labeling systems and for the scientists wishing to use them. I think the work will be of interest to many in various fields and is suitable for publication in Nat. Comm. following some revision. The authors may also want to make some edits to make sure their findings are more apparent to a broad audience of RNA-tag users--not just single molecule and RNA folding specialists. For example, I'd like to suggest that the authors articulate more clearly the significance of their findings in the Abstract.

Major Comments

1. I think the authors may be a bit too negative on the apparent mechanical instability of Spinach. In some cases, this could be beneficial. In particular, mechanical instability may aid in allowing the spinach tag to be efficiently degraded by the cell. The authors may wish to bring this up, particularly in light of work by Singer and Parker (Nat Methods. 2018 Jan; 15(1):81-89).

2. I think an outstanding question the authors are in the position to answer is "how many of the RNA mango or spinach aptamers are capable of becoming fluorescent"? Do all molecules of the aptamer on the surface exhibit DFHBI or T01 fluorescence when the ligands are added? I think this would be a very interesting value to report--analogous to the "maturation" percentage of GFP. As far as I know, there is no information about this for either aptamer.

3. I think the authors may want to give a bit more background on Mango. Early iterations of Mango required an interaction between the RNA and the biotin moiety of T01-biotin for tight binding. I believe this interaction is removed in iMango III and MangoIV but this may warrant some clarification/explanation.

4 I found the folding trajectories in 5c to be interesting but was hoping to see more examples to get information on the homogeneity of the transitions. Is it possible to synchronize the trajectories and create heat-map like plots of the transitions? This would be similar to what several labs studying translation by smFRET have done (see for example Blanchard et al., PNAS, 2004, pg 12893, Fig. 4).

Related to these experiments, can error bars be included for the fit parameters shown in Figure S14c? Can fit methods be included in the experimental? I found the fit and kinetic data related to these to be a bit under-described in the text. Is it possible to experimentally validate that base triple formation and stem zipping are responsible for the GQ-only intermediates?

Minor Comments:

1. Line 97-- "enables"

2. Can the authors explain either in the main text or figure legend, how the CD spectra in the supplemental data (e.g., Figure S1b) confirms GQ formation? For those of us who aren't GQ aficionados, the spectra all resemble one another.

3. I found the sentence on lines 169-170 a bit confusing. It seems to indicate that interactions are present in E1 and E3 but not E2. Maybe an alternate interpretation is that there are no changes in contacts b/n DFHBI and RNA that result in altered folding upon entry or exit from E2?

Reviewer #2 (Remarks to the Author):

This manuscript describes the folding and unfolding dynamics of "light-up" RNA aptamers, Spinach2, iMangoIII and MangoIV. The method is single-molecule fluorescence-force spectroscopy with smFRET and optical tweezers. The authors investigated how some aspects of the folding and

unfolding kinetics are similar and different among the three cases in the presence of K^+ and Mg^{2+} ions. The results indicate that the folding/unfolding kinetics of an aptamer depends on various parameters including mono- and di-valent ions and an external force stretching the RNA molecule. The dependence is specific to each aptamer, suggesting that different aptamers may result in different results in quantitative cellular imaging.

The conclusions are well supported by the results. The conclusions and results will be useful to interpret imaging data obtained with these aptamers. I support their publication with some minor comments as follow that I hope the authors can address in a revision.

1. I do not understand the description of the significance of the results in lines 190-193. I suggest the authors be more elaborative and realistic about when an unfolding mechanical force to an aptamer is relevant in a cell. Their example of RNA polymerase is particularly strange. They need to give more realistic circumstances when an aptamer could be under such a force and what the implications would be in quantitative imaging experiments.

2. It is unclear why 1 mM Mg^{2+} was used for MangoIV while 5 mM was used for the other two aptamers. Without any justification of these different levels of Mg^{2+} , the authors concluded that MangoIV folding is not sensitive to Mg^{2+} while Spinach2 folding is. I suggest the authors add some explanations/justifications.

3. Related to comment #2, 5 mM Mg^{2+} seems a bit too high as many reported that free Mg^{2+} in eukaryotic cells is at 1-2 mM. Is 5 mM good to generate a pronounced change within a short incubation time? Or is there any other reason? I suggest the authors add some explanations on this and to revise the description in lines 322-324 accordingly.

Response to Reviewers' comments

Title: Nanomechanics and co-transcriptional folding of Spinach and Mango

The authors greatly appreciate the comments and suggestions from the reviewers, incorporation of which have improved the manuscript. Our response to each comment appears in blue.

Reviewer #1 (Remarks to the Author):

Mitra and Ha have carried out a thorough and well-conducted study of two different types of RNA aptamer. They study folding of the aptamers using smFRET and through creative experiments using a combined fluorescence-force setup and a "co-transcriptional-like" system. The results are interesting and provide food for thought for the chemical biology community interested in developing these RNA labeling systems and for the scientists wishing to use them. I think the work will be of interest to many in various fields and is suitable for publication in Nat. Comm. following some revision. The authors may also want to make some edits to make sure their findings are more apparent to a broad audience of RNA-tag users--not just single molecule and RNA folding specialists. For example, I'd like to suggest that the authors articulate more clearly the significance of their findings in the Abstract.

Major Comments

1. I think the authors may be a bit too negative on the apparent mechanical instability of Spinach. In some cases, this could be beneficial. In particular, mechanical instability may aid in allowing the spinach tag to be efficiently degraded by the cell. The authors may wish to bring this up, particularly in light of work by Singer and Parker (Nat Methods. 2018 Jan;15(1):81-89).

Response: We greatly thank the reviewer for bringing this point to our notice. We have added the following to the Discussion section:

“However, mechanical instability of Spinach may facilitate efficient degradation of Spinach-tagged mRNA, thereby rendering it an optimal tool for study of instantaneous number of mRNA with short life spans.”

2. I think an outstanding question the authors are in the position to answer is "how many of the RNA mango or spinach aptamers are capable of becoming fluorescent"? Do all molecules of the aptamer on the surface exhibit DFHBI or TO1 fluorescence when the ligands are added? I think this would be a very interesting value to report--analogous to the "maturation" percentage of GFP. As far as I know, there is no information about this for either aptamer.

Response: We recorded DFHBI-1T-induced fluorescence in ~ 39 % of the Spinach2 molecules (Figure S1f). We added the following to the Results section

“Approximately 39 % of the Spinach2 molecules tethered on the single molecule surface manifested GFP-like fluorescence upon addition of DFHBI-1T (Figure S1f). Maturation

propensities of wt-GFP and EYFP are respectively ~ 50 % and ~ 75 %, at 32 °C. However, in view of the ultrafast dissociation kinetics of DFHBI from Spinach under continuous laser illumination used in our study, a significant population of fluorescent complexes may have been inadvertently missed at our time resolution of 200 ms.”

TO1 binding to the Mango aptamers were examined by fluorescence imaging of Cy3-labelled Mango, while TO1 was tethered on the surface via biotin-neutravidin interactions. We were unable to observe direct fluorescence activation of TO1-Mango complex at the single molecule level likely due to incompatibility of our laser/emission filter combination with that required for detection of Mango fluorescence.

3. I think the authors may want to give a bit more background on Mango. Early iterations of Mango required an interaction between the RNA and the biotin moiety of T01-biotin for tight binding. I believe this interaction is removed in iMangoIII and MangoIV but this may warrant some clarification/explanation.

Response: We have added an introductory statement on Mango aptamers as follows:

“Since the first report of engineered Mango aptamer by Dolgosheina et al, Mango variants with improved fluorescent properties, binding affinities etc have been identified via functional reselection.”

Although TO1-Biotin was used for reselection of Mango aptamers via SELEX, any interaction between the biotin moiety of TO1-Biotin and the reselected Mango aptamer (MangoII, MangoIII and MangoIV), or the lack of it, has not been elucidated in the crystal structures.

4. I found the folding trajectories in 5c to be interesting but was hoping to see more examples to get information on the homogeneity of the transitions. Is it possible to synchronize the trajectories and create heat-map like plots of the transitions? This would be similar to what several labs studying translation by smFRET have done (see for example Blanchard et al., PNAS, 2004, pg 12893, Figure4).

Related to these experiments, can error bars be included for the fit parameters shown in Figure S14c? Can fit methods be included in the experimental? I found the fit and kinetic data related to these to be a bit under-described in the text. Is it possible to experimentally validate that base triple formation and stem zipping are responsible for the GQ-only intermediates?

Response: We thank the reviewer for the suggestions. We have now included heat-map like plots of the single molecule trajectories in Figure 5c. We have also included a brief description of GQ formation/dwell time fit method in the Materials and Method section and the corresponding error estimates in Figure S14c.

We designated the vectorial folding intermediates of iMangoIII and MangoIV as folded GQs, based on comparative FRET values obtained under the refolding format.

In Figs.3 and S5, we have demonstrated that base triple formation in *iMangoIII* is initiated only in the presence of Mg^{2+} . However, as Mg^{2+} is essential for ATP hydrolysis and unwinding activity of Rep-X, we were unable to trap *iMangoIII* in its GQ-only conformation in the vectorial folding experiment. Nevertheless, Mg^{2+} alone induces ssRNA-like behavior (Figure S9) with $E \sim 0.4$ which is different from the $E \sim 0.55$ intermediate obtained under vectorial folding conditions, making it unlikely that a ssRNA-like structure is responsible for the $E \sim 0.55$ intermediate.

MangoIV manifested cooperative folding of the GQ-core and the duplex stem in our refolding assays. Thus, under the near physiological conditions used in the vectorial folding assay, we could not capture independent GQ folding or stem zipping events.

We hope to address formation of GQs or base triplex/duplex stem as intermediates to mature *Mango* aptamer folding in future experiments.

Minor Comments:

1. Line 97-- "enables"

Response: The grammar has been corrected in the manuscript.

2. Can the authors explain either in the main text or figure legend, how the CD spectra in the supplemental data (e.g. Figure S1b) confirms GQ formation? For those of us who aren't GQ afficionados, the spectra all resemble one another.

Response: We have added the signature spectral features of parallel GQs in the Results section of the main text.

3. I found the sentence on lines 169-170 a bit confusing. It seems to indicate that interactions are present in E1 and E3 but not E2. Maybe an alternate interpretation is that there are no changes in contacts b/n DFHBI and RNA that result in altered folding upon entry or exit from E2?

Response: The authors thank the reviewer for his insightful comment. We acknowledge this possibility and have rephrased our hypothesis as follows:

“These changes in the Δx_u^\ddagger s may be suggestive of force-induced changes in interactions between DFHBI-1T and Spinach2 motifs during E_1 and E_3 unfolding but not during E_2 unfolding.”

Reviewer #2 (Remarks to the Author):

This manuscript describes the folding and unfolding dynamics of “light-up” RNA aptamers, Spinach2, *iMangoIII* and *MangoIV*. The method is single-molecule fluorescence-force spectroscopy with smFRET and optical tweezers. The authors investigated how some aspects of the folding and unfolding kinetics are similar and different among the three cases in the presence of K^+ and Mg^{2+} ions. The results indicate that the folding/unfolding kinetics of an aptamer depends on various parameters including mono- and di-valent ions and an external force

stretching the RNA molecule. The dependence is specific to each aptamer, suggesting that different aptamers may result in different results in quantitative cellular imaging.

The conclusions are well supported by the results. The conclusions and results will be useful to interpret imaging data obtained with these aptamers. I support their publication with some minor comments as follow that I hope the authors can address in a revision.

1. I do not understand the description of the significance of the results in lines 190-193. I suggest the authors be more elaborative and realistic about when an unfolding mechanical force to an aptamer is relevant in a cell. Their example of RNA polymerase is particularly strange. They need to give more realistic circumstances when an aptamer could be under such a force and what the implications would be in quantitative imaging experiments.

Response: We have rephrased our significance statement in the section stated above as follows:

“Such low force threshold, lower than the forces exerted by RNA motor proteins such as helicases and other RNA binding proteins may lead to inadvertent unfolding of the aptamer and compromise fluorogen binding and hence “light-up” properties of Spinach2 in cellular imaging applications.”

2. It is unclear why 1 mM Mg^{2+} was used for MangoIV while 5 mM was used for the other two aptamers. Without any justification of these different levels of Mg^{2+} , the authors concluded that MangoIV folding is not sensitive to Mg^{2+} while Spinach2 folding is. I suggest the authors add some explanations/justifications.

Response: *iMangoIII* and Spinach2 manifested Mg^{2+} induced structural transitions pertaining to formation and stabilization of base triplet junctions adjacent to the respective GQ cores. We have provided a hypothesis in the manuscript as follows:

Spinach2: “ Mg^{2+} intercalates between the top G-quartet and the U_{32} - A_{64} - U_{61} base triple at the junction with the P3 stem. As the P3 stem constitutes the loop connecting Gs across all the G-quartets (Figure S1a), we propose the formation of partially folded GQs ($E \sim 0.7$) in 100 mM K^+ alone, which evolve into a stably folded form ($E \sim 0.86$) upon addition of 5 mM Mg^{2+} (Figure S4e).”

iMangoIII: “In the absence of force, K^+ -induced folding of *iMangoIII* gave a smFRET population at $E \sim 0.55$ (Figures 3a, c (left), S5b (left)) with a CD spectral signature corresponding to parallel GQs (Figure S5a). Interestingly, on supplementing the buffer with 5 mM Mg^{2+} , we observed an additional structural evolution to a stable state with $E \sim 0.74$ (Figures 3b, c (right) and S5b (right)). Because the crystallographic structure of *iMangoIII* showed that divalent cations coordinate base triple formation beneath the G-quartet, we attribute the E transition from ~ 0.55 to ~ 0.74 to the stacking between A_7 , G_{19} and U_{26} (Figures 3a and b).”

Hence, for *iMangoIII* and *Spinach2*, we performed our measurements in buffers containing 100 mM K^+ and 5 mM Mg^{2+} , to ensure stable formation of the respective GQ cores and base triplets.

For *MangoIV*, we have provided the following description:

“Although the above zero-force smFRET and fluorescence-force spectroscopy measurements indicated cooperative folding of the GQ core and duplex stem of *MangoIV* in a buffer containing 100 mM K^+ only, we repeated the measurements in a buffer containing 1 mM Mg^{2+} in addition to 100 mM K^+ to better mimic the intracellular ionic conditions.”

3. Related to comment #2, 5 mM Mg^{2+} seems a bit too high as many reported that free Mg^{2+} in eukaryotic cells is at 1-2 mM. Is 5 mM good to generate a pronounced change within a short incubation time? Or is there any other reason? I suggest the authors add some explanations on this and to revise the description in lines 322-324 accordingly.

Response: Our results suggest Mg^{2+} dependent stabilization of base triplets in *iMangoIII* and *Spinach2* such that stable base triplet folding was observed at ~ 5 mM Mg^{2+} . Both *iMangoIII* and *Spinach2* demonstrated E transitions ($E_{0.55}$ to $E_{0.74}$ for *iMangoIII*, Figure S5c and $E_{0.7}$ to $E_{0.86}$ for *Spinach2*, Figure S4b and c) close to the concentration of free Mg^{2+} (1-2 mM), in eukaryotic cells. We hypothesized partially folded GQs corresponding to the $E_{0.7}$ state in *Spinach2*, wherein fluorogen binding is significantly compromised (Figure S4d). Thus, under physiologically relevant Mg^+ concentrations, optimal “light-up” activity cannot be realized in *Spinach2*. We have clarified this point in lines 330-331 of the present version of manuscript. Impact of Mg^+ dependence on *Spinach*-DFHBI fluorescence has been addressed in Refs. 15 and 46.

On the other hand, as GQ formation in *iMangoIII* is independent of the base triplet, its “light-up” properties are relatively insensitive to the concentrations of Mg^{2+} (Figure S8a and f).

Regarding the response time to changes in magnesium concentration: All zero-force smFRET measurements were performed within a minute of addition of imaging buffer. Significant differences in FRET histograms collected under varying salt conditions (Figures S1, S4 (*Spinach2*), S5 (*iMangoIII*) and S10 (*MangoIV*)), underscore fast response times of the aptamers to changes in cationic concentrations.

REVIEWERS' COMMENTS:

Reviewer #1 (Remarks to the Author):

In their revised manuscript, Mitra and Ha have thoroughly responded to reviewer comments and the manuscript has been improved. I recommend acceptance and publication with only a very minor correction.

Minor Corrections:

1. Line 327, since this is a bit theoretical at this stage, I would avoid calling Spinach an "optimal tool". Perhaps just "thereby rendering it a potentially useful tool for accurately..."